# Microbial Selection and Functional Adaptation in Technical Snow: A Molecular Perspective from 16S rRNA Profiling

**DOI:** 10.3390/ijms26199712

**Published:** 2025-10-06

**Authors:** Anna Lenart-Boroń, Piotr Boroń, Bartłomiej Grad, Klaudia Bulanda, Natalia Czernecka-Borchowiec, Anna Ratajewicz, Klaudia Stankiewicz

**Affiliations:** 1Department of Microbiology and Biomonitoring, Faculty of Agriculture and Economics, University of Agriculture in Krakow, Mickiewicza Ave. 24/28, 30-059 Krakow, Poland; 2Department of Forest Ecosystem Protection, Faculty of Forestry, University of Agriculture in Krakow, 29 Listopada Ave. 46, 31-425 Krakow, Poland; piotr.boron@urk.edu.pl (P.B.); bartlomiej.grad@urk.edu.pl (B.G.);; 3Scientific Circle of Biotechnologists, Faculty of Biotechnology and Horticulture, University of Agriculture in Krakow, 29 Listopada Ave. 54, 31-425 Krakow, Poland; natalia.czernecka2@student.urk.edu.pl (N.C.-B.); anna.ratajewicz@student.urk.edu.pl (A.R.)

**Keywords:** 16S rRNA sequencing, bacterial community structure, cold-adapted bacteria, environmental filtering, functional prediction, ski resorts, storage reservoirs, technical snow

## Abstract

Artificial (technical) snow production is an increasingly common practice in alpine regions, yet little is known about its role in shaping microbial communities at the molecular level. In this study, we combined culture-based methods with high-throughput 16S rRNA gene sequencing and functional trait prediction (FAPROTAX) to investigate bacterial communities across the full technical snowmaking cycle in one of Polish ski resorts. The molecular profiling revealed that technical snow harbors dominant taxa with known cold-adaptation mechanisms, biofilm-forming abilities, and stress tolerance traits (e.g., *Brevundimonas*, *Lapillicoccus*, *Massilia*, with a relative abundance of 2.95, 2.14, 3.38 and 5.61%, respectively). Functional inference revealed a consistent dominance of chemoheterotrophy (up to 38% in relative abundance) and aerobic chemoheterotrophy (up to 36%), with localized enrichment of fermentation (6.9% in cannon filter and 6.5% in sediment) and aromatic compound degradation (3.7% in source waters, 3.8% in cannon filter and 4.6% in sediment). Opportunistic and potentially pathogenic genera (e.g., *Acinetobacter*, *Flavobacterium*, *Nocardia*) persisted in sediments (7.4%, 21.4% and 3.5%) and meltwater (34.9% and 2.31% for the latter two), raising concerns about their environmental reintroduction. Our findings indicate that technical snowmaking systems act as selective environments not only for microbial survival but also for the persistence of molecular traits relevant to environmental resilience and potential pathogenicity. Our study provides a molecular ecological framework for assessing the impacts of snowmaking on alpine ecosystems and underscores the importance of monitoring microbial functions in addition to taxonomic composition.

## 1. Introduction

In recent years, artificial snowmaking has become an indispensable component of ski resort operations across Europe, including Poland. As natural snowfall becomes increasingly unpredictable, the reliance on technical snow—produced by combining water and compressed air—has grown substantially [1]. To meet this demand, snowmaking systems utilize various water resources, such as rivers, streams, different types of reservoirs and occasionally treated wastewater. These water sources are often rich in microorganisms and can serve as microbial vectors into the snowmaking infrastructure and ultimately onto ski slopes [2,3]. Previous studies have shown that water used for technical snow production harbors diverse microbial communities, including potentially pathogenic and antibiotic-resistant bacteria [4,5]. The microbial quality of source waters, especially when reclaimed water or surface runoff is used, poses ecological and public health concerns, as snowmelt may return microorganisms to downstream aquatic ecosystems and/or soil [6].

Snow, regardless of origin, represents a climatically sensitive and transient ecosystem that links the atmosphere with the underlying soils, influences the hydrological cycle and interacts with diverse ecosystems. These characteristics apply to both natural and technical (i.e., ‘artificial’) snow. However, technical snow differs from natural snow in its higher density [7], which prolongs melting and may intensify its influence on the surrounding aquatic environments near ski slopes where it is produced and stored. Snowpacks are known to harbor distinct microbial communities, often shaped by environmental origin and physicochemical conditions [8,9,10]. Regardless of the snow type, it creates harsh conditions with limited liquid water and extreme temperatures, creating a habitat with a specific “core” microbiome [11]. Snow creates a specialized environment for bacterial communities, with technical snow forming an even more unique ecological niche. Harsh conditions that prevail in snow drive molecular-level adaptations such as the expression of cold-shock proteins, antifreeze proteins, and membrane fluidity regulators, enabling microbial persistence in snow environments. All the above can promote the development of a specialized microbiota, the composition of which may differ significantly not only from that of natural snow but also from that of the microbial communities present in the water used to feed snow cannons. Importantly, snow—especially of technical origin—may act both as a barrier and a vector for microbial dissemination, with potential implications for downstream environmental microbiology and human exposure.

Although the physicochemical properties of technical snow have been relatively well studied [10,12,13,14], little is known about how snowmaking systems affect microbial community structures and functions across the full production cycle. In particular, the molecular mechanisms underlying microbial survival, selection, and adaptation during snow production, as well as the potential for microbial dissemination through snow and meltwater, remain poorly understood. Few studies have combined culture-based and high-throughput sequencing approaches to examine these dynamics.

Recent advances in high-throughput sequencing, especially 16S rRNA gene amplicon profiling, enable detailed taxonomic and functional characterization of microbial communities in engineered and natural environments [15]. Functional prediction tools like FAPROTAX further allow inference of metabolic traits, including chemoheterotrophy, fermentation, and aromatic compound degradation, based on taxonomic profiles [16]. These molecular insights are essential to understand how snowmaking systems shape microbial community structure and function, and how these changes may impact downstream ecosystems [17]. To date, most microbial research in cold environments has focused on natural snow, glaciers or polar systems [8,11,18,19], whereas artificially produced snow remains underexplored.

This study aimed to characterize bacterial communities across the full technical snow production system in a Polish ski resort, using a dual approach of culture-based bacterial isolation and high-throughput 16S rRNA gene amplicon sequencing. Sampling sites included feed water sources, storage reservoirs, snowmaking infrastructure (e.g., snow cannon filters), and different stages of technical snow life (freshly produced, aged, and meltwater), as well as downstream sediments. Functional profiles were inferred using FAPROTAX to identify key microbial processes associated with snow microbiomes. To our knowledge, this is the first comprehensive microbiological and molecular assessment of the full artificial snow production and transformation cycle. The findings provide a comprehensive molecular perspective on microbial selection, redistribution and functional transformation in technical snow, with potential implications for microbial transport and ecological processes in downstream aquatic systems.

## 2. Results and Discussion

To assess how microbial communities change across stages of technical snow production, we integrated culture-based enumeration of bacteria with high-throughput 16S rRNA gene sequencing to obtain information about total bacterial communities across all sample types. These analyses were supplemented with functional prediction via FAPROTAX. This dual approach enabled us to capture both viable and non-culturable bacterial fractions and infer metabolic traits relevant to cold adaptation, stress tolerance, and ecological function.

### 2.1. Abundance of Culturable Bacteria and OTU Richness Across the Snow Production Pipeline

Figure 1 shows total CFU (sum of *E. coli*, enterococci, coagulase-positive staphylococci, and mesophilic bacteria) as blue bars (log scale, left axis), while OTU counts based on 16S rRNA gene amplicon sequencing are shown as a red line (right axis). Detailed CFU data for individual bacterial groups are provided in Appendix A. The data (Figure 1, Appendix A) illustrate the contrast between culture-based and sequencing-based detection, particularly in snow and meltwater samples, where OTUs remain detectable despite low or absent culturable bacteria. This discrepancy reflects the limitations of culture-based methods, which typically capture less than 1% of environmental bacterial diversity [20,21]. Also, attempts to correlate culture-based CFU counts of indicator organisms (e.g., *E. coli*, *Enterococci* or *Staphylococci*) with their relative abundances in the 16S rRNA gene amplicon data revealed no consistent patterns, reflecting fundamental methodological differences between culture-dependent and culture-independent approaches and the potential presence of DNA from non-viable cells. It also supports the value of high-throughput molecular methods such as 16S rRNA sequencing in uncovering the microbial diversity, including viable but non-culturable (VBNC) and dead cells retaining DNA signatures in low-biomass, oligotrophic or stressed environments, like snow. Snow cannon filters and sediment samples exhibit the highest microbial loads in both approaches, highlighting their potential as microbial reservoirs or biofilm-rich sites within the system. Presumably, snow cannons may trap organic debris and provide surfaces for microbial attachment, as well as provide shelter from direct UV light exposure, which is favorable for microbial persistence. Similar observations have been reported in engineered aquatic environments, where biofilm accumulation and particle trapping contribute to increased microbial biomass and diversity [22]. In contrast, the near absence of culturable bacteria in fresh snow and snowmelt suggests harsh environmental elimination, possibly due to freezing, UV exposure, or nutrient scarcity. These factors inhibit the growth of non-adapted taxa but do not preclude their molecular detection, which captures DNA from viable but non-culturable cells and dead cells as well [4,11,19]. This may explain why OTUs are still observed at sites where CFUs are nearly absent.

### 2.2. Microbial Diversity and Richness

Bacterial diversity varied clearly across the stages of technical snow production (Table 1). The highest richness and diversity were observed at the WWTP and Intake sites, reflecting the complexity and heterogeneity typical of anthropogenically impacted freshwater systems. This observation is consistent with earlier findings that such environments act as microbial hubs, integrating a wide range of bacterial taxa from diverse sources [4,23].

In contrast, reservoirs showed reduced diversity and evenness, likely as a result of environmental filtering and passive treatment processes such as UV exposure or settling [4]. Both fresh snow samples exhibited low diversity and high dominance, suggesting strong selective pressure during snow formation. Factors such as high-pressure spraying, rapid freezing, and low nutrient content likely favor psychrotolerant or stress-adapted taxa while excluding others [11,24,25]. Interestingly, the cannon filter B and reservoir sediment samples retained relatively high diversity and richness, consistent with their role as microbial accumulation zones or biofilm reservoirs within the system [26,27]. Their protected surfaces and stable conditions may promote colonization and retention of a broader microbial community.

To evaluate compositional similarities between samples, we calculated Bray–Curtis dissimilarities based on the 50 most abundant genera across all samples (Figure 2). The calculated dissimilarity values are very high, suggesting clear variation in bacterial community composition across the technical snowmaking system. However, the highest similarities were observed between WWTP and Intake (dissimilarity = 0.26), two snow cannons A1 and A2 (0.47) and between the two reservoirs (0.44). Cannon A1 also showed moderate similarity to reservoir sediment B (0.39). These patterns suggest that spatial proximity, similar environmental conditions, or shared functions (e.g., anthropogenically impacted water, water retention, biofilm formation and freezing temperatures) can support overlapping microbial assemblages. In contrast, snow-related samples (Fresh Snow A, Fresh Snow B and Aged Snow A) showed high or nearly complete dissimilarity with source waters (e.g., Fresh Snow A vs. WWTP = 0.97; Fresh Snow A vs. Intake = 0.95; Fresh snow B vs. WWTP = 0.86), underscoring the strong selective pressure during the snow formation processes. These patterns align with previous studies showing that technical and environmental filtering, including UV light disinfection, pressurized spraying and water freezing, significantly alter microbial communities in engineered water systems [28,29,30,31]. Overall, these results support the model of progressive microbial selection and restructuring from input waters through snowmaking and deposition. Also the fact that snow and melt samples cluster away from input water sources, support the idea that cold adaptation, desiccation tolerance, and low nutrient availability drive microbial divergence in snow environments [32,33]. Meanwhile, sediment samples retained greater microbial similarity to each other and to cannon filters, likely due to their role as microbial sieves, enriched in particle-associated and biofilm-forming bacteria [34].

### 2.3. Bacterial Community Composition

The composition of bacterial communities across the technical snow production system revealed pronounced shifts in dominant taxa, as visualized in Figure 3 and Figure 4. At the phylum level (Figure 3), the most abundant groups across all samples were Proteobacteria, Bacteroidota, Actinobacteriota, and Firmicutes, although their proportions varied substantially by site. Proteobacteria clearly dominated in reservoir samples, where their relative abundance exceeded 50%. They also consistently prevailed in cannon filters, freshly produced technical snow and sediment samples. Snowmelt water, however, was dominated by Bacteroidota (45.3%), with Proteobacteria as the second most abundant phylum (31.5%), indicating a shift in bacterial community composition during thawing.

Aged snow displayed a more balanced community composition, with co-dominance of Proteobacteria (45.8%), Actinobacteriota (30.2%), and Bacteroidota (22.6%). These patterns suggest selective survival or accumulation effects within the snowpack over time. In WWTP and Intake water samples, Proteobacteria and Actinobacteriota were most abundant, consistent with their prevalence in urban and freshwater environments. Meanwhile, cannon filters showed high proportions of Bacteroidota and Proteobacteria, while Firmicutes became co-dominant in the Cannon filter B (27.5%), possibly indicating niche adaptation or biofilm formation in response to operational conditions. Finally, sediment samples (A and B) showed mixed phylum-level composition, with notable variation between the two reservoirs.

At the order level (Figure 4), bacterial community composition was even more variable across sample types and individual samples. Flavobacteriales dominated the snow continuum, being most abundant in Fresh snow (41.8%), Aged snow (18.1%) and Snowmelt (35.6%) samples, suggesting adaptability of their members (such as these found in Antarctic soils like *F. antarcticum*, *F. glaciei*, *F. sinopsychrotolerans*, etc. [35]) to cold and/or oligotrophic aquatic environments. Such adaptation strategies include cold-shock proteins, ice-binding proteins, glycogen and proline formation, maintaining permeability and fluidity of the cellular membrane [36]. They also dominated in snow cannon filters A, reinforcing their role as key snow-associated taxa. Burkholderiales, a diverse order commonly associated with both freshwater and soil environments, was also highly abundant across multiple sites, including reservoirs, cannon filters, snow and sediments. Surprisingly, Rhizobiales, indicative of soil or plant origin, were highly prevalent in Snow B (63.1%), suggesting airborne, soil or plant-related microbial inputs [27,33] into the snowpack or the source waters (in this case reservoir B, where the relative abundance of Rhizobiales was nearly 5%). Notably, the proportion of Rhizobiales in reservoir A and snow A was much lower (around 1%), suggesting that the local surroundings of source water may influence the dominant bacterial community composition in downstream samples. Corynebacteriales were the most abundant in the Outflow A sample (32.6%) and the second most abundant in Aged snow A (15.2%), suggesting either their persistence or enrichment over time [10,37]. In contrast, the WWTP sample exhibited the highest abundance of Enterobacterales, consistent with their association with anthropogenic sources. Interestingly, Cannon filters A1 and A2 retained detectable levels of Enterobacterales (1.5% and 3.4%, respectively), and they were also present in Outflow A (2.3%) and Reservoir B sediment (2.1%), suggesting partial retention or downstream transport and deposition of wastewater-associated taxa within the system.

These taxonomic shifts likely reflect a combination of environmental filtering (e.g., freezing and thawing, desiccation) treatment effects (e.g., UV light disinfection, aeration in reservoirs) coupled with surface-associated microbial selection in snow cannon filters or in bottom sediments of reservoirs [26,27,38]. All these factors shape the microbial communities involved in technical snow production and transformation, and during snowmelt, they may further influence the composition of microbial communities in receiving waters and underlying soil [25,39].

### 2.4. Ecologically Distinct Taxa and Indicators Across Snow Production Stages

One of the aims of this study was to profile bacterial community shifts throughout the five-month cycle of technical snow production and transformation. Several taxa emerged as ecologically or functionally distinct, serving as indicators of particular environments or selective pressures along the process. While the overall taxonomic composition varied across the water-to-snow-to-sediment pathway, the presence of certain genera provided insight into their possible source, selective pressure put thereon and bacterial adaptations to changing conditions.

Table 2 and Figure 5 summarize the key genera selected based on their abundance patterns, ecological significance, and known environmental niches. Analysis of their abundance within the examined anthropogenically impacted environment offered a deeper understanding of microbial dynamics and possible effects on further elements of the receiving compartments. Wastewater-associated bacteria such as Candidatus *Microthrix* and unidentified genera of *Enterobacteriaceae* family were most abundant by the wastewater treatment plant (WWTP), but nearly absent at downstream sites, indicating their dilution or removal during subsequent stages. The river water intake for technical snowmaking (Intake site) showed a community typical of moderate anthropogenic impact, including *Nocardia* (9.36%) or potentially pathogenic freshwater opportunists *Acinetobacter* (5.12%) and biofilm-forming *Flavobacterium* (7.80%) [24]. Technical reservoirs, which are the subsequent step in technical snow production, appeared to facilitate both attenuation of anthropogenic indicators and enrichment of resilient taxa such as *Comamonas*, *Massilia* [40] or *Flavobacterium* [24], suggesting their dual role as both biological filters and selective habitats for cold-adapted bacteria.

Snow cannon filters, another step in technical snow production, selected strongly for surface-associated, biofilm-forming or stress-tolerant taxa. These included not only members of the *Xanthomonadaceae* family, *Flavobacterium*, and *Brevundimonas*, but also taxa adapted to harsh conditions, like *Massilia* (early colonizers of oligotrophic habitats, isolated from glacial meltwater [40,41]) or anaerobic *Tepidibacter* [42]. Likely, the filters provide surfaces with periodic water flow, nutrient scarcity and physical trapping, which on one hand allow for the purification of contaminated water and on the other—shape the downstream bacterial community. Freshly produced technical snow showed simple microbial communities, dominated by *Flavobacterium* (snow A) or *Rhizobium* cluster (snow B). It suggests that physical stress of snow production, i.e., high-pressure water spraying, air mixing, freezing) favors specific groups, likely these with fast response to shock and rapid surface attachment ability [24,33]. The ageing of snow on ski slopes is reflected by the community composition observed in samples such as Aged snow A and Snowmelt A, where the bacterial community diversified. In these samples, *Flavobacterium* and *Comamonas* remained abundant, but additional taxa with distinct habitat preferences started to appear. These included *Rhizobium* (soil or plant origin), *Brevundimonas* (oligotrophic environments), and *Lapillicoccus*, typically isolated from harsh, nutrient-poor extreme environments, such as Antarctic soils after long incubation periods [43]. Cold-adapted genera like *Sphingomonas* [11,32,41], and anaerobes commonly found in anoxic environments such as *Pelosinus* [44] were also present, reflecting post-depositional ecological shift and interaction with surrounding environment.

Another aspect to consider relevant in terms of microbial input into the environment as a result of technical snow production, is the fact that changing the ecological balance never remains without consequences for downstream ecosystems. Several taxa identified in the samples most likely to contribute to the microbial composition downstream (i.e., snowmelt water, reservoir outflow and sediments) show moderate to high relevance in terms of pathogenic potential to humans, animals or plants, and may have important ecological impact. *Acinetobacter*, prevalent in Sediment A, includes species known as nosocomial pathogens with strong biofilm-forming ability and resistance to disinfection, posing a potential risk if released during reservoir maintenance or flushing [45,46]. Environmental taxa, like *Comamonas* (highly prevalent in Sediment A and Snowmelt water) or *Brevundimonas* (enriched in Outflow A and Sediment B), adapt to nutrient-poor environments, contribute to biofilm formation and include species recently reported as important opportunistic pathogens [47,48]. Members of the genus *Dickeya*, known phytopathogens, and *Flavobacterium*, which includes opportunistic pathogens to humans and fish and species with algicidal activity, may affect soil or aquatic health if introduced into vegetated zones or irrigation systems [49,50,51]. Two taxa within the *Clostridium sensu stricto* complex were detected in multiple samples. *Clostridium sensu stricto* 1 includes well-known human and animal pathogens, such as *C. perfringens* and *C. botulinum*, capable of forming resistant spores and surviving harsh conditions, posing a latent risk upon environmental release [52]. *Clostridium sensu stricto* 13, while less characterized, is associated with fermentative activity in sediments and anaerobic digesters, and may signal eutrophic or anoxic conditions [53]. The persistence of these *Clostridium* groups in cannon filters and sediments raises concerns about long-term microbial dissemination, particularly during snowmelt or sediment displacement. These findings underline the importance of monitoring microbial communities not only for direct pathogenic threats but also for their environmental consequences following snowmelt or sediment dispersal.

### 2.5. Functional Prediction of Bacterial Communities Inferred by FAPROTAX

To infer the metabolic potential of bacterial communities, 16S rRNA-based taxonomic profiles were analyzed using the FAPROTAX database. This approach allowed us to assign putative ecological functions based on the known metabolism traits.

The most prevalent functional categories across all compartments were chemoheterotrophy and aerobic chemoheterotrophy, consistent with heterotrophic bacterial dominance (Figure 6 and Figure 7, Appendix A). These functions were particularly enriched in snow-associated samples such as Snow A (0.377 and 0.364, respectively) and Aged Snow A (0.346 and 0.335), followed by Snowmelt A (0.339 and 0.326), indicating the presence of metabolically active, cold-adapted heterotrophs capable of surviving harsh, nutrient-limited conditions. Aromatic compound degradation-associated taxa were detected in most compartments (Figure 6) but were especially enriched in Sediment A (0.046), snow cannon filter A2 (0.038), and STP (0.037), suggesting possible retention or accumulation of pollutant-degrading taxa within infrastructure components. Fermentation was among the less abundant but environmentally relevant functions. Fermentative potential was the highest in cannon filter B (0.069) and sediment (0.065) (Appendix A, Figure 7) samples, suggesting that these compartments may be characterized by low-oxygen or anoxic conditions, promoting anaerobic metabolic processes [54]. In contrast, surface water and fresh snow samples showed significantly lower values (below 0.015).

Interestingly, the predicted functional profiles revealed contrasts between the sampling sites. Snow A exhibited low number (only nine) of distinct functional groups, dominated by chemoheterotrophy and aerobic respiration, indicating a low-complexity microbial community shaped by strong selection pressures during snow production. In contrast, Sediment A and Cannon Filter B showed much higher functional richness (31 and 26 functions, respectively; Appendix A), reflecting the capacity of these environments to accumulate and sustain metabolically diverse bacterial populations. On the other hand, Snow B had the highest relative abundance of taxa classified as “others” by FAPROTAX (68%), suggesting a prevalence of taxa not well-represented in current databases or possessing poorly annotated metabolic capabilities. This divergence may indicate microbial input from alternative sources (e.g., aerosols, biofilms), differential infrastructure contamination, or variability in snowmaking processes. Taken together, these findings highlight the heterogeneity of technical snow environments and their distinct roles in shaping microbial functional potential.

Overall, the FAPROTAX results revealed clear stage-specific functional profiles, indicating that snowmaking systems not only shape taxonomic community structure but also modulate microbial metabolic potential with implications for downstream biogeochemical cycling.

### 2.6. Principal Component Analysis of Microbial Communities

To further explore the dynamics in bacterial community composition, we conducted principal component analysis (PCA) on the relative abundance of dominant genera. The first three components explained 48.5% of the total variance (Figure 8A,B).

PC1 (explaining 22.95% of variance) reflected variation in anaerobic, sludge or sediment-associated communities with highest contributions from *Paludibacter*, *Sporomusa* and *Ruminiclostridium* (members of Clostridia). These are anaerobic bacteria, typically dwelling in sediments or biofilm-associated environments (in our study: snow cannons and sediment) [55,56].

PC2 (13.90% of variance) separated samples influenced by biofilm-forming and cold-adapted taxa (i.e., Aged snow with taxa such as *Sphingomonas*, *Dyadobacter*) [11,22] from those dominated by treatment-resistant organisms such as Candidatus *Microthrix* and *Caldilineaceae* (samples WWTP, Intake, Outflow), suggesting a filtering effect during snow production and water treatment processes.

Finally, PC3 explained 11.60% of variance and reflected transitions possibly linked to site-specific microbial adaptations that occur in time. The effect presented by PC3 is mostly influenced by *Nocardia*, *Rudanella* and *Paenibacillus*, soil-associated, organic matter-degrading taxa [57,58]. The genus *Paenibacillus* has also been reported to dwell in nutrient-poor conditions, a trait that is shared with *Pedobacter*, which also significantly contributed to PC3 and is typically described as a psychrotolerant and oligotrophic bacterium [59,60].

The patterns described above are consistent with the suggestions we proposed earlier in this section, as well as with other studies that emphasize the selective pressure exerted on bacterial communities by technical processes and environmental conditions (including UV light disinfection, pressurized water spraying and freezing) in engineered systems [4,61]. They also underscore the ecological relevance of cold-tolerant and oligotrophic taxa in snow and glacier-like environments [11,32].

## 3. Materials and Methods

### 3.1. Site Description and Sample Collection

The collection of samples was conducted in one of the Polish ski resorts in the Polish Carpathians. Due to a non-disclosure agreement with the facility operators, the exact location cannot be provided. The snowmaking infrastructure at this site includes water intake from a mountain river, water storage in technical reservoirs, and a snow cannon network distributed across ski slopes.

A total of fourteen samples were collected before, during and after the 2023/2024 snowmaking season (i.e., from November 2023 to April 2024) to represent the full spectrum of the technical snow production and transformation process. The samples included are shown in Table 3.

In all cases, the samples were collected in three instantaneous replications that formed the final pooled sample. This resulted in formation of a single analytical sample per site that was subjected to further experiments. The samples of water and sediment were collected into sets of autoclaved 1000 mL sterile polyethylene bottles, while snow samples were collected by first scratching the superficial layer, followed by the collection of snow with a snow corer (a 1.0 m-long, 10 cm-wide tube) into double sterile plastic string bags, where it melted. Then snowmelt water was transferred into sets of 1000 mL sterile polypropylene bottles and analyzed. The snow cannon biofilm and sediment was collected by scratching with sterile spatulas and swabs. The collected material was placed in 100 mL sterile Falcon tubes. Sediments were homogenized prior to processing.

### 3.2. Culture-Based Microbial Analysis of Samples

To assess the abundance of viable and potentially health-relevant bacteria, four microbial groups were enumerated in all samples: *Escherichia coli*, fecal enterococci (*Enterococcus faecalis*/*E. faecium*), coagulase-positive staphylococci and total mesophilic bacteria. The following culture media were used for each group: Tryptone Bile X-glucuronide (TBX) agar (Biomaxima, Lublin, Poland) for *E. coli*, Slanetz-Bartley agar (Biomaxima, Lublin, Poland) for enterococci, Baird-Parker agar (Biomaxima, Lublin, Poland) for staphylococci and Tryptic Soy Agar (TSA, Biomaxima, Lublin, Poland)) for total mesophilic heterotrophic bacteria.

For water and snowmelt water samples, *E. coli* and enterococci were quantified using the membrane filtration method, with 100 mL of each sample filtered through a 0.22 µm membrane filter (Sartorius, Germany) and transferred onto the respective media. Plates were incubated at 37 °C for enterococci and 44 °C for *E. coli* according to standard microbiological protocols. Staphylococci and heterotrophic mesophilic bacteria in liquid samples were assessed by the pour plate method, in which 1 mL of each sample was plated into sterile Petri dishes and overlaid with the appropriate medium. Incubation was carried out at 37 °C for 24–48 h depending on the target group. Solid samples, including reservoir sediments and biofilms from snow cannon filters, were analyzed by preparing serial tenfold dilutions from 1 g of homogenized material in sterile saline solution. Subsequently, 1 mL of each dilution was plated onto the respective media using the pour plate technique.

Colonies characteristic of each group were counted, and the results are expressed as the numbers of colony forming units (CFU) per ml or 100 mL of liquid samples or per gram for solid samples. All microbiological procedures were conducted under aseptic conditions to avoid cross-contamination.

### 3.3. DNA Extraction and 16S rRNA Sequencing

For molecular analysis of the total bacterial community, genomic DNA was extracted from all samples using standard protocols optimized for environmental matrices. For liquid samples (water and snowmelt), a volume of 500 mL was filtered through 0.22 µm sterile membrane filter (Sartorius, Germany). Filters were immediately transferred to sterile 60 mm Petri plates and stored at −20 °C until DNA extraction. For solid samples (sediment and snow cannon biofilm), approximately 0.5 g of material was used for each extraction. Nuclease-free water (A&A Biotechnology, Gdańsk, Poland) was used as a DNA extraction blank and was further processed alongside the examined samples.

DNA extraction was performed using the Genomic Mini AX Bacteria + extraction kit (A&A Biotechnology, Gdańsk, Poland), following the manufacturer’s protocol. DNA was purified using Anty-Inhibitor Kit (A&A Biotechnology, Gdańsk, Poland) and DNA concentration was measured fluorometrically on a Qbit 4 Fluorometer (ThermoFisher Scientific, Waltham, WA, USA). Amplicon libraries targeting the hypervariable V3–V4 region of the 16S rRNA gene were prepared following the Illumina 16S Metagenomic Sequencing Library Preparation Guide (Part #15044223 Rev. B, Illumina, San Diego, CA, USA). A two-step PCR protocol was used for amplification, employing the following primers: forward primer: 5′-TCGTCGGCAGCGTCAGATGTGTATAAGAGACAGCCTACGGGNGGCWGCAG-3′ and reverse primer 5′-GTCTCGTGGGCTCGGAGATGTGTATAAGAGACAGGACTACHVGGGTATCTAATCC-3′ [62]. PCR reactions were performed using Herculase II Fusion DNA Polymerase and indexed with the Nextera XT Index Kit v2 (Agilent Technologies, Santa Clara, CA, USA). PCR mix without a DNA template was used as a negative control (also called no-template control). The resulting libraries were quality-checked and pooled before sequencing. Sequencing was conducted on the Illumina MiSeq platform (2 × 300 bp paired-end) at Macrogen Inc. (Seoul, Republic of Korea).

### 3.4. Bioinformatics and Statistical Methods

The normality of data was assessed using the Shapiro–Wilk test. Since the distributions were approximately normal, parametric tests were applied in subsequent analyses. Differences in bacterial counts as well as functional traits between sample types and among individual samples were evaluated using one-way analysis of variance (ANOVA), followed by post hoc comparisons using the least significant difference (LSD) method. A significance threshold of *p* < 0.05 was applied in all statistical tests.

Principal component analysis (PCA) was used to investigate relationships between culturable bacterial counts and the relative abundance of bacterial genera identified through Illumina sequencing. The number of components retained was determined using the Kaiser criterion, with factors exhibiting eigenvalues greater than 1.0 considered for interpretation. All statistical analyses were performed using Statistica software, version 13 (TIBCO Software Inc., Palo Alto, CA, USA).

Sequencing data targeting the V3–V4 region of the 16S rRNA gene were taxonomically classified by aligning reads against the Greengenes database (version 13; 97% similarity threshold, minimum score 40). Sequences were clustered into operational taxonomic units (OTUs) and assigned to taxonomic levels down to genus using the QIIME2 pipeline. To assess differences in bacterial community composition across samples, Bray–Curtis dissimilarity was calculated using relative abundance data of bacterial taxa. The resulting pairwise dissimilarity matrix was used to evaluate beta diversity between sample types. A clustered heatmap was generated with hierarchical clustering applied to rows (samples) based on Bray–Curtis distances. The relative abundance of dominant bacterial phyla and genera was calculated and visualized using R software (version 4.4.2) with the packages: vegan [63], ggplot2 [64] and pheatmap [65].

Functional prediction was performed using the FAPROTAX database (Functional Annotation of Prokaryotic Taxa) as implemented in the Python script version [v.1.2.4]) [16]. The normalized OTU table obtained after taxonomic assignment was used as a input. Functional annotations were inferred based on taxonomic affiliation and matched to ecological functions. The output was used to generate relative abundance plots to visualize predicted metabolic potentials across the sampling stages.

## 4. Conclusions

This study provides the first comprehensive molecular-level characterization of microbial communities across the full cycle of technical snow production and transformation from water intake to snowmelt water and post-season reservoir sediments. By integrating high-throughput 16S rRNA gene sequencing with functional prediction using FAPROTAX, we demonstrated that each stage of the snowmaking system—from treated wastewater that contaminates river intake, through reservoirs and snow cannons to snow deposited on the slopes and sediments that accumulate throughout the season—acts as a distinct ecological and functional filter, progressively shaping the microbial communities and metabolic potential.

Our results highlight the dual role of technical snowmaking infrastructure. On the one hand, it can reduce microbial richness and create a selective environment that reduces many anthropogenic bacteria, but on the other hand, it is a temporary reservoir for biofilm-forming, cold-adapted and stress-tolerant taxa. Functional predictions revealed consistent dominance of aerobic chemoheterotrophic processes, while specialized functions such as methanol oxidation and fermentation were confined to specific compartments like sediments and cannon filters. Importantly, certain genera identified in sediments and snowmelt (e.g., *Acinetobacter*, *Comamonas*, *Clostridium*, *Flavobacterium*) may possess both environmental relevance and potential pathogenic or resistance-associated traits, highlighting the need for further genomic investigation.

Given that snowmelt, reservoir outflow and sediments may reintroduce these microbes into natural or semi-natural environments, our findings underscore the importance of molecular monitoring of microbial community dynamics and functional potentials not only from a public health perspective but also to understand possible long-term impacts of technical snow production processes on aquatic ecosystems.

Future studies could consider seasonal variation, antibiotic resistance genes, and virulence factors and compare natural versus artificial snow systems to better assess the ecological and public health consequences of using technical snowmaking in ski resorts and other engineered environments.

## Figures and Tables

**Figure 1 ijms-26-09712-f001:**
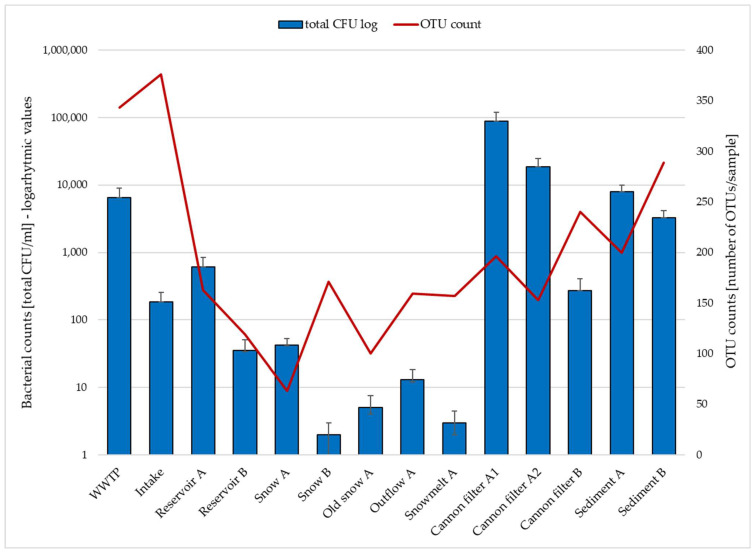
Comparison of total culturable bacterial counts (CFU/mL) and OTU richness (numbers of OTU per sample) across various sampling sites involved in technical snow production.

**Figure 2 ijms-26-09712-f002:**
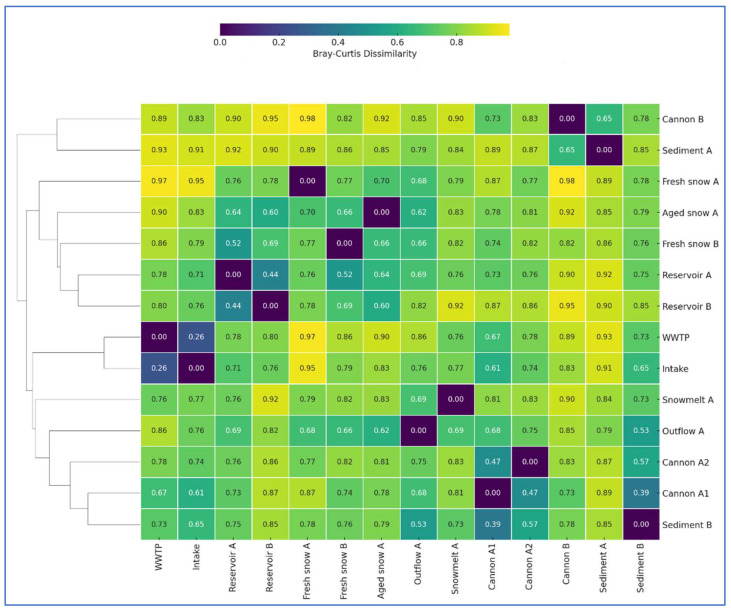
Clustered heatmap of Bray–Curtis dissimilarity values among bacterial across all sampling sites in the technical snow production system. Bray–Curtis dissimilarities were calculated based on relative abundance data of 50 most abundant bacterial taxa. Lower values (purple) indicate higher community similarity, while higher values (yellow) represent greater dissimilarity. Hierarchical clustering was applied to rows (samples) to reveal community structure groupings.

**Figure 3 ijms-26-09712-f003:**
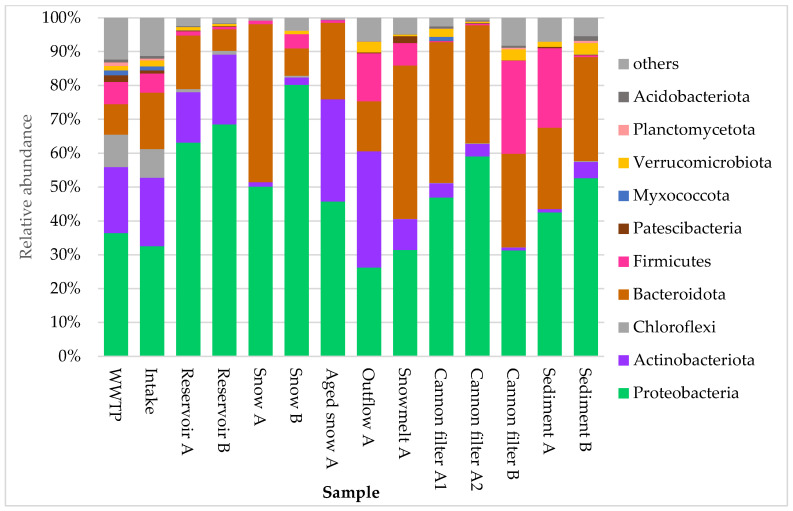
Relative abundance of ten dominant bacterial phyla in samples collected from various stages of technical snow production and the related aquatic systems. Each bar represents the taxonomic composition at the phylum level, based on 16S rRNA gene sequencing, expressed as percentage of total reads per sample. The graph presents ten most abundant taxa, while “others” represent the group of less abundant ones.

**Figure 4 ijms-26-09712-f004:**
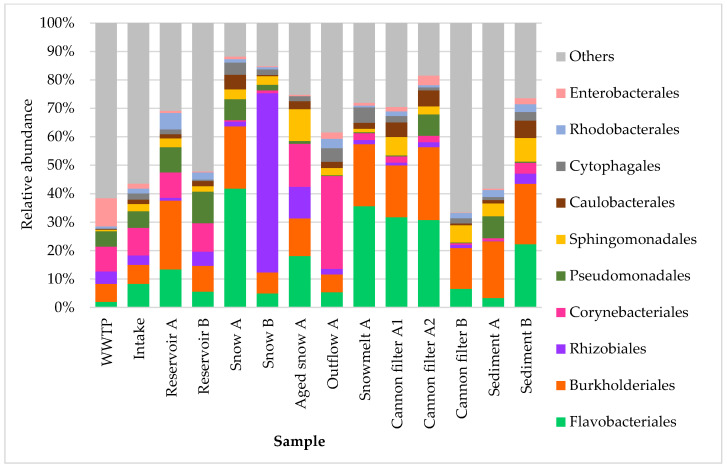
Relative abundance of ten dominant bacterial orders in samples collected from various stages of technical snow production and the related aquatic systems. Each bar represents the taxonomic composition at the order level based on 16S rRNA gene sequencing, expressed as percentage of total reads per sample. The graph presents ten most abundant taxa, while “others” represent the group of less abundant ones.

**Figure 5 ijms-26-09712-f005:**
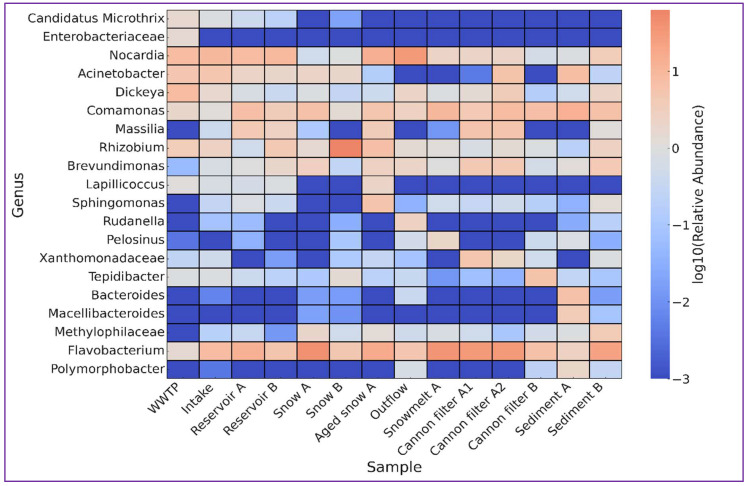
Heatmap showing the log_10_-transformed relative abundance of selected bacterial taxa across different sampling sites involved in the technical snow production cycle. The analyzed taxa represent ecologically or functionally distinct groups, including these typical of wastewater (e.g., Candidatus *Microthrix*, *Enterobacteriaceae*), cold-adapted or oligotrophic bacteria (e.g., *Sphingomonas*, *Lapillicoccus*), and genera associated with sediments or biofilms (e.g., *Comamonas*, *Brevundimonas*). The scale bar indicates the relative abundance values after logarithmic transformation. Sampling sites include treated wastewater discharge site (WWTP), river water intake for technical snowmaking, water storage reservoirs (A and B), snow cannon filters (A1, A2, B), fresh and aged snow samples, snowmelt, and post-season sediments.

**Figure 6 ijms-26-09712-f006:**
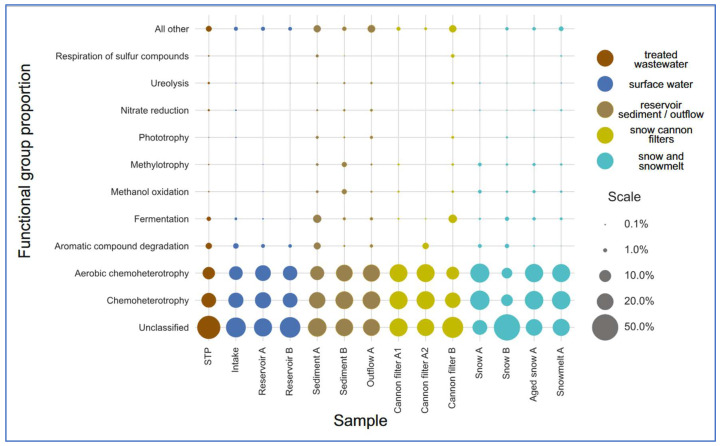
Relative abundance of predicted bacterial functional groups across sample categories, based on FAPROTAX analysis. Samples were grouped into functional stages of technical snow production: wastewater treatment plant (STP), intake water, reservoirs, snow (fresh, aged), snowmelt, cannon filters, outflow, and sediments. The most abundant functions include chemoheterotrophy and aerobic chemoheterotrophy. The category “other” includes less specific or unassigned functions.

**Figure 7 ijms-26-09712-f007:**
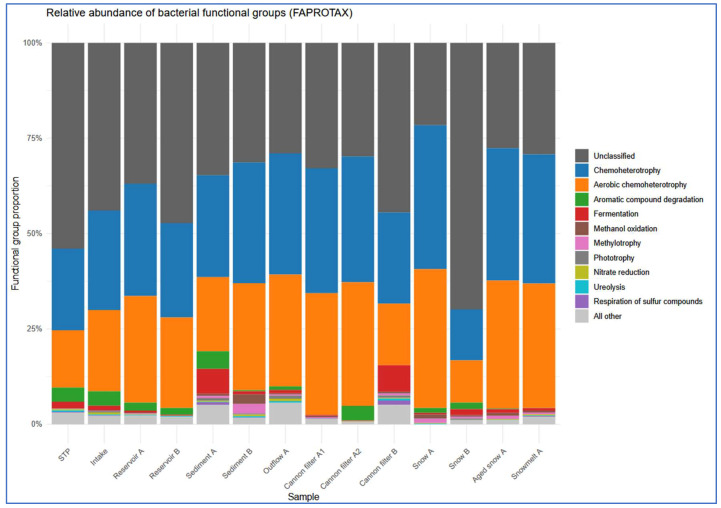
Site-specific distribution of ten most prevalent FAPROTAX-predicted functions. Each bar represents a single sampling site, showing the relative contribution of the most dominant metabolic functions.

**Figure 8 ijms-26-09712-f008:**
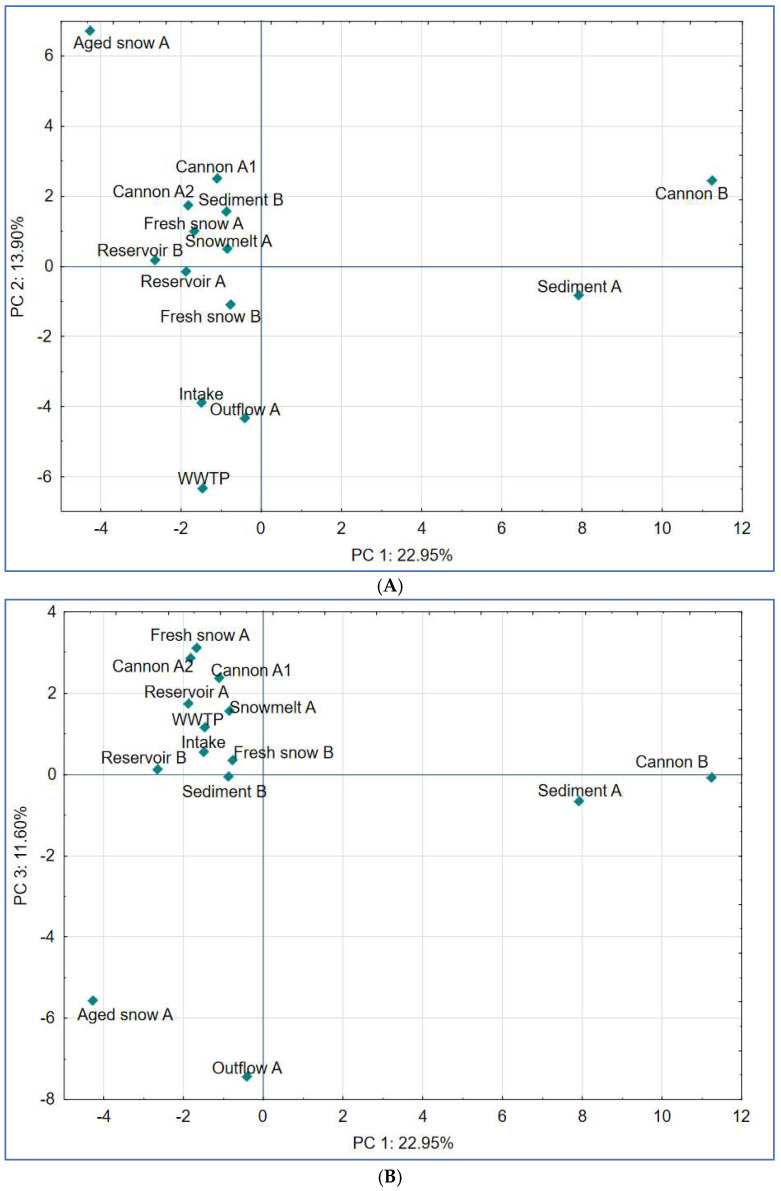
Results of principal component analysis (PCA) based on bacterial community composition at the genus or the closest level across all sampled stages of the examined technical snow production system. (**A**) PCA plot showing separation along principal components PC 1 and PC 2, which explain 22.95% and 13.90% of the total variance, respectively; (**B**) PCA plot showing separation along principal components PC 1 and PC 3, explaining 22.95% and 11.60% of the total variance, respectively.

**Table 1 ijms-26-09712-t001:** Diversity indices calculated for the bacterial community composition across samples collected throughout the technical snow production system. The indices include Shannon diversity (H), Simpson diversity (Ds), Simpson’s dominance (λ), Pielou diversity (PIE) and Pielou evenness (J), Margalef richness (Dm) and observed species richness (number of OTUs). A heatmap-style coloring scheme was used, with shades of blue indicating lower values and shades of red indicating higher values for each diversity metric.

	H	Ds	λ	PIE	J	Dm	Species Richness
WWTP	1.753	0.961	0.039	0.961	0.709	62.682	296
Intake	1.833	0.967	0.033	0.967	0.731	68.267	321
Reservoir A	1.402	0.927	0.073	0.927	0.646	31.775	148
Reservoir B	1.211	0.851	0.149	0.851	0.592	24.424	111
Fresh snow A	1.086	0.804	0.196	0.804	0.603	13.755	63
Fresh snow B	0.878	0.598	0.402	0.598	0.399	33.808	159
Old snow A	1.367	0.926	0.074	0.926	0.690	21.013	96
Outflow	1.403	0.886	0.114	0.886	0.649	31.197	145
Snowmelt	1.300	0.856	0.144	0.856	0.597	32.467	151
Cannon filter A1	1.415	0.887	0.113	0.887	0.628	38.265	179
Cannon filter A2	1.288	0.883	0.117	0.883	0.599	29.954	141
Cannon filter B	1.831	0.974	0.026	0.974	0.792	43.557	205
Sediment A	1.688	0.959	0.041	0.959	0.757	36.213	169
sediment B	1.592	0.934	0.066	0.934	0.667	50.499	243

**Table 2 ijms-26-09712-t002:** Key bacterial taxa characteristic of individual snow production and post-production stages.

Genus	Typical Habitat	Ecological/Functional Remarks	Sampling Site of Highest Relative Abundance	Max. Observed Abundance
Candidatus *Microthrix*	activated sludge systems; wastewater treatment plants	associated with sludge bulking and foaming issues	WWTP	1.67
*Enterobacteriaceae*	wastewater, wastewater treatment plants	typical indicators of wastewater contamination	WWTP	1.58
*Nocardia*	soil, water, nutrient-poor environments	survives nutrient-poor conditions, opportunistic pathogens	Outflow Reservoir A	31.05
*Acinetobacter*	various environments	includes potential pathogens, biofilm forming	Sediment A	7.43
*Dickeya*	wastewater, wastewater treatment plants	environmental bacterium, frequent plant pathogen	WWTP	7.95
*Comamonas*	carbon-poor aquitard sediments	adaptability to nutrient-limited environments, opportunistic pathogen	Sediment A	13.51
*Massilia*	glacial meltwater	early colonizers in the rhizosphere	Snow cannon filter A2	5.80
*Rhizobium*	agricultural runoff, plants	biofilm formation, nitrogen fixing	Snow B	62.91
*Brevundimonas*	oligotrophic environments	biofilm formation, opportunistic pathogen	Snow cannon filter A2	4.51
*Lapillicoccus*	harsh, nutrient-poor, extreme environments	isolated from stone surfaces; aerobic and mesophilic organisms	Aged snow A	2.14
*Sphingomonas*	glacier cryoconite, Arctic environments	psychrophilic, cold-adapted, increases plant resistance to pathogens	Aged snow A	5.61
*Rudanella*	activated sludge systems	potentially involved in organic matter degradation	Outflow reservoir A	2.74
*Pelosinus*	subsurface and sedimentary environments	anaerobic	Snowmelt A	2.02
*Xanthomonadaceae*	biofilm, contaminated source water	biofilm formation	Snow cannon filter A1	4.99
*Tepidibacter*	anaerobic conditions	anaerobic	Snow cannon filter B	6.08
*Bacteroides*	wildlife feces	fecal indicator	Sediment reservoir A	6.18
*Macellibacteroides*	wildlife feces	fecal indicator	Sediment reservoir A	3.88
*Clostridium sensu stricto 1* and *13*	soils, organic rich sediments,	spore-forming, can survive disinfection, includes pathogens	Snowmelt/Sediment reservoir A	1.21/2.4
*Methylophilaceae*	stratified water bodies	methylotrophic bacteria	Sediment reservoir B	3.84
*Flavobacterium*	environments rich in organic substrates; water	biofilm formation, fish pathogen, algicidal, biopolymer decomposer, opportunistic pathogen	Snow A	41.55
*Polymorphobacter*	cold, oligotrophic environments	adaptability to extreme conditions	Sediment A	2.00

**Table 3 ijms-26-09712-t003:** The samples collected for the analysis.

No	Sample Characteristics	Sample Code
Water sources		
1	Treated wastewater outflow from a local municipal treatment plant	WWTP
2	River water from the main snowmaking intake	Intake
3	Water from two technical reservoirs used to store water prior to snow production; Reservoirs are equipped with water aeration and UV light disinfection systems	Reservoir A
4		Reservoir B
Snow production system		
5	Biofilm or debris from three snow cannon filters	Cannon A1
6		Cannon A2
7		Cannon B
8	Two samples of freshly produced technical snow	Fresh snow A
9		Fresh snow B
Post-deposition samples		
10	Aged snow collected from the ski slope A c.a. 2 months post deposition (February)	Aged snow A
11	Snowmelt water from aged snow patches, collected in April	Snowmelt
12	Reservoir (A) water outflow collected at the end of the snowmaking season (April)	Outflow A
13	Sediments from the bottom of both technical reservoirs, exposed after post-season drainage of water	Sediment A
14		Sediment B

## Data Availability

Sequencing data were deposited on NCBI and are publicly available under the BioProject number: PRJNA1336629. The remaining data are available on request from the corresponding author.

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
