# Peer review of "Microbial Selection and Functional Adaptation in Technical Snow: A Molecular Perspective from 16S rRNA Profiling"

_ijms, 2025, doi:10.3390/ijms26199712_

Round 1
Reviewer 1 Report
Comments and Suggestions for Authors
The study was written well and had new idea. So, i accept for publication in this journal.
Author Response
Comment 1: The study was written well and had new idea. So, i accept for publication in this journal.
Response: Thank you very much for the appreciation of the quality of our manuscript. We tried to do our best.
Reviewer 2 Report
Comments and Suggestions for Authors
This study provides a comprehensive molecular-level characterization of microbial communities across the technical snow production cycle, offering valuable insights into microbial selection and functional adaptation in these unique, engineered environments. The dual-approach, combining culture-based methods with high-throughput sequencing, is a major strength, and the analysis logically progresses from community structure to functional potential. The findings are significant for understanding the ecological impacts of snowmaking. However, several major and minor issues need to be addressed before this manuscript is suitable for publication.
Major issues:
- In the abstract section, I find no key data. For example, the authors mentioned distinct microbial assemblages; however, they did not provide the relative abundance percentages of dominant genera or the approximate proportion of key functions (e.g., chemoheterotrophy) in the samples.
- The introduction provides a clear rationale for the study. To further improve its flow and focus, the discussion of the research gap, which appears in multiple places (e.g., paragraphs 1, 3, and 5), could be streamlined and expressed more concisely.
- Lines 278-280: The authors mention the high abundance of Rhizobiales (63.1%) in the Snow B sample, which is interesting. Furthermore, they proposed a hypothesis of airborne, soil, or plant-related inputs. However, the authors do not explain why Snow A lacks this pattern. I suggest that they address this point to clarify the contrast.
- In lines 270-273, the text attributes the dominance of Flavobacteriales to “their adaptability to cold and/or oligotrophic aquatic environments.” However, the authors do not explain why Flavobacteriales can thrive in cold environments.
- The references format is inconsistent;please check the manuscript. For instance, in line 163, “Kumar et al. 2018)” is used, while other references follow a numerical format.
- Lines 506-507: The Methods section states that for each site, samples were collected in "three instantaneous replications that formed the final pooled sample". It is important for the authors to clarify that this approach results in a single analytical sample per site, not independent biological replicates.
- A critical omission in the Methods section is the lack of any mention of negative controls, such as DNA extraction blanks (processed alongside the samples) and no-template controls (NTCs) for the PCR steps.
Minor issues:
- Line 25: The word “chemocheterotrophy” is misspelled. It should be corrected to ‘chemoheterotrophy’.
- Line 27: The word “reflected” should be changed to the present tense “reflect.”
- Line 130: There is a missing space between “16S” and “rRNA.”
- Line 160: The word “hideout” would be better changed to “shelter.”
- Lines 163-168: The sentence is too long and contains two “which” clauses. It should be revised for clarity.
- In line 320, an article “the” should be added before “subsequent step.”
- Line 416: “326” should be changed to “0.326.”
- Line 472: The phrase “has been also reported” should be revised to “has also been reported.”
Author Response
Comment 1: In the abstract section, I find no key data. For example, the authors mentioned distinct microbial assemblages; however, they did not provide the relative abundance percentages of dominant genera or the approximate proportion of key functions (e.g., chemoheterotrophy) in the samples.
Response 1: As suggested the Abstract section was supplemented with key data. However, in order to fit within the required limit of words, we needed to alter the text slightly. We hope that the Abstract is acceptable in this form.
Comment 2: The introduction provides a clear rationale for the study. To further improve its flow and focus, the discussion of the research gap, which appears in multiple places (e.g., paragraphs 1, 3, and 5), could be streamlined and expressed more concisely.
Response 2: We agree with the Reviewer and we reorganized the Introduction so that the research gap is stated concisely in one paragraph and that the individual paragraphs refer to different aspects, and do not repeat the same information.
Comment 3: Lines 278-280: The authors mention the high abundance of Rhizobiales (63.1%) in the Snow B sample, which is interesting. Furthermore, they proposed a hypothesis of airborne, soil, or plant-related inputs. However, the authors do not explain why Snow A lacks this pattern. I suggest that they address this point to clarify the contrast.
Response 3: Thank you for this remark, we completely agree with it. As additionally suggested by the Editor, the reservoir B also contained higher proportion of Rhizobiales than reservoir A. Therefore, we referred to both these probabilities, i.e. the fact that the source water contained different amounts of bacteria detected in snow and the fact that the surroundings of the reservoir may quite importantly influenced the dominant bacterial community composition.
Comment 4: In lines 270-273, the text attributes the dominance of Flavobacteriales to “their adaptability to cold and/or oligotrophic aquatic environments.” However, the authors do not explain why Flavobacteriales can thrive in cold environments.
Response 4: We added an explanation of the mechanisms by which some members of Flavobacteriales (mostly Flavobacterium species) can thrive in cold environments and provided relevant references.
Comment 5: The references format is inconsistent;please check the manuscript. For instance, in line 163, “Kumar et al. 2018)” is used, while other references follow a numerical format.
Response 5: Thank you for pointing this out. We have checked and corrected it. This mistake has occurred due to the fact that we used Mendeley citation add-in and there must have been when the Bibliography was inserted.
Comment 6: Lines 506-507: The Methods section states that for each site, samples were collected in "three instantaneous replications that formed the final pooled sample". It is important for the authors to clarify that this approach results in a single analytical sample per site, not independent biological replicates.
Response 6: Thank you for this comment. We added this information as suggested.
Comment 7: A critical omission in the Methods section is the lack of any mention of negative controls, such as DNA extraction blanks (processed alongside the samples) and no-template controls (NTCs) for the PCR steps.
Response 7: Thank you for this remark. Of course, negative controls were used throughout the process. Firstly, negative control was used alongside the samples in the DNA extraction protocol and then was subjected to the DNA quality check prior to PCR reactions. Then, no-template control was applied in order to prepare the samples for further processing in Macrogen, Europe (Illumina sequencing). We have added this information into the text in relevant places.
Comment 8: all minor issues:
Line 25: The word “chemocheterotrophy” is misspelled. It should be corrected to ‘chemoheterotrophy’.
Corrected
Line 27: The word “reflected” should be changed to the present tense “reflect.”
This word has now been removed, as the Abstract text was altered a bit to meet the recommendations of the Reviewer.
Line 130: There is a missing space between “16S” and “rRNA.”
corrected
Line 160: The word “hideout” would be better changed to “shelter.”
corrected
Lines 163-168: The sentence is too long and contains two “which” clauses. It should be revised for clarity.
corrected
In line 320, an article “the” should be added before “subsequent step.”
done
Line 416: “326” should be changed to “0.326.”
cone
Line 472: The phrase “has been also reported” should be revised to “has also been reported.”
done
Also the manuscript was once again read and looked through to search for the spelling and grammatical errors. We hope that all of them are now corrected.
Reviewer 3 Report
Comments and Suggestions for Authors
The paper entitled “Microbial Selection and Functional Adaptation in Technical Snow: A Molecular Perspective from 16S rRNA Profiling”, authors are Anna Lenart-Boroń, Piotr Boroń, Bartłomiej Grad, Klaudia Bulanda, Natalia Czernecka, Anna Ratajewicz and Klaudia Stankiewicz, is an original work focused on studies of microbial diversity and its changes in the technical snow used in ski resorts. This is the first work devoted to the microorganisms related with the process of technical snowmaking and its melting. The paper shows the influence of this process and the snow itself on the environment contacting with artificial snow. The originality and novelty are evident. Both methods of cultivation / bacterial isolation and genetic analysis were used, in contrast to many works based on molecular biology only. This is a very good point because it better helps to understand how microbial communities respond to human interferences. Moreover, an additional analysis using FAPROTAX was performed to estimate trophic functions of microorganisms. This all makes the study to be a proper microbiological work. The statistical analysis was appropriate with suitable numbers of samples, trials, replicates and controls. The up-to-date literature was used for references.
Specific comments
- Figure 1 – It is strongly recommended to add labels for axes with units and statistical approvements (e.g. SD or p-values).
Did authors synchronize (compare / align) data of 16S rRNA metagenomics and cultured groups of microorganisms? Do they confirm each other in numbers or ratios? The question is because of, if the number of CFU of a certain microbial group (E. coli, enterococci, staphylococci, or total mesophilic bacteria) is high (like 100000 cells per ml or more), this should strongly impact on abundance of specific taxa. For example, number of staphylococci will impact on abundance of Bacillota (former Firmicutes), and the number of E. coli + enterococci will impact on abundance of Pseudomonadota (former Proteobacteria). Is it possible to make some predictions of a ratio of DNA from dead cells in amplicon analysis on the base of numbers of cultivating microorganisms?
Author Response
Comment 1: Figure 1 – It is strongly recommended to add labels for axes with units and statistical approvements (e.g. SD or p-values)
Response 1: Thank you for this comment. Indeed, the figure was unclear in that form. The labels with units for axes were added as suggested and standard deviation was added to the CFU values. Due to the fact that the number of OTUs was a single value per sample, no standard deviation measure could have been calculated.
Comment 2: Did authors synchronize (compare / align) data of 16S rRNA metagenomics and cultured groups of microorganisms? Do they confirm each other in numbers or ratios? The question is because of, if the number of CFU of a certain microbial group (E. coli, enterococci, staphylococci, or total mesophilic bacteria) is high (like 100000 cells per ml or more), this should strongly impact on abundance of specific taxa. For example, number of staphylococci will impact on abundance of Bacillota (former Firmicutes), and the number of E. coli + enterococci will impact on abundance of Pseudomonadota (former Proteobacteria). Is it possible to make some predictions of a ratio of DNA from dead cells in amplicon analysis on the base of numbers of cultivating microorganisms?
Response 2:
Thank you for this valuable comment. We attempted to correlate the two types of data (i.e. CFU of E. coli, enterococci and staphylococci with the relative abundances of corresponding taxa in 16S rRNA gene amplicon data) in some other analyzes that we now conduct and also in the current dataset, following the Reviewer’s question. No consistent relationships were observed – samples with high CFU counts of, for example, E. coli, did not show a higher relative abundance of Proteobacteria or of Enterobacterales in the sequencing results. For this reason, we think that it is not possible to make predictions of such ratio in amplicon analysis based on the number of cultivated microorganisms.
This apparent discrepancy likely reflects several well-known factors: Firstly: different target units: CFUs enumerate viable cells able to grow on a given medium, while 16S amplicon sequencing quantifies total bacterial DNA, including non-culturable or dead cells. Secondly, PCR and primer biases and differences in 16S rRNA gene copy numbers among taxa can distort relative abundance estimates. Also, DNA from dead cells may persist and be amplified even when CFU are low, and conversely some living cells may be under-represented if they are difficult to lyse or if their 16S gene copies are few. And finally, the absolute abundance of a single group that reaches 10⁵ CFU/ml is still a small fraction of the total microbial community in these samples, so its effect on relative percentages can be minor. For these reasons, predicting the ratio of DNA from dead cells based on CFU counts is not currently feasible. We have now added a brief note in the Discussion to clarify this point.
What also needs to be mentioned here is the fact that E. coli belong to Enterobacteriaceae family and Proteobacteria phylum, while Enterococcus is Firmicutes, not Proteobacteria.
Round 2
Reviewer 2 Report
Comments and Suggestions for Authors
The revision successfully addresses all of the earlier comments. With both major and minor issues satisfactorily resolved, the manuscript is suitable for acceptance.